# Analysis of Agricultural Drought Risk Based on Information Distribution and Diffusion Methods in the Main Grain Production Areas of China

**Kaijie Niu [1,2], Qingfang Hu [2], Lu Zhao [1], Shouzheng Jiang [1], Haiying Yu [1], Chuan Liang [1,*] and Yintang Wang [2,*]**

[1] State Key Laboratory of Hydraulics and Mountain River Engineering & College of Water Resource and Hydropower, Sichuan University, Chengdu 610065, China; nkj3596@163.com (K.N.); luya1121@163.com (L.Z.); shouzhengjiang@sina.com (S.J.); yuhaiying0722@sicnu.edu.cn (H.Y.)
[2] Nanjing Hydraulic Research Institute, Nanjing 210029, China; hqf_work@163.com
[*] Correspondence: lchester@sohu.com (C.L.); ytwang@nhri.cn (Y.W.)

**Abstract:** Accurate assessment of agricultural drought risk is of strategic significance to ensure future grain production security in the main grain production areas of China. Agricultural drought risk assessment is based on drought vulnerability characteristics. In this study, firstly the drought thresholds were redefined by correlation analysis of drought strength based on the Standardized Precipitation Evapotranspiration Index (SPEI) and drought damage rates, then the information distribution and the two-dimensional normal information diffusion method were employed to establish the vulnerability curve between drought strength and drought damage rates. Finally, provincial drought risks and the conditional probabilities at different drought damage stages were obtained. The results show that the drought vulnerability curve was nonlinear. With the increase of drought strength, drought damage rates increased rapidly at the beginning, and after a small fluctuation locally, they no longer increased significantly and tended to be relative stable. The occurrence probabilities of agricultural drought risk presented great spatial differences, with the characteristics of high in the northern, moderate in the central and southwestern part, and lower in the southeastern provinces in the main grain production areas of China. The analysis of conditional probability showed that Hubei, Henan, and Jiangxi were the provinces most prone to drought-affected risk under the drought-induced condition; while Liaoning, Hunan, and Inner Mongolia were the ones most prone to lost harvest risk under the drought-induced or the drought-affected condition. The results could be used to provide guidance for drought risk management and to formulate appropriate plans by the relevant departments.

**Keywords:** agricultural drought; vulnerability; risk assessment; Standardized Precipitation Evapotranspiration Index (SPEI); information distribution; information diffusion; China

## 1. Introduction

Being one of the most reported natural disasters of the last decades, drought often causes severe damages to society, the natural ecosystems, and the economy [1–3]. Economic losses due to drought are far exceeding other natural disasters almost every year [4]. With rapid population growth, increasing water demand, and limited water supplies, drought is doubtlessly becoming more and more severe and frequent [5]. China is a country vulnerable to drought disaster, which is causing an impediment to sustaining the development of the national economy and social stability [6]. As China is a great agricultural country, the impact of droughts on agricultural yields has gained more and more attention in recent years. The Chinese government has indicated that ensuring the country's food supply is the top priority of all national work, especially in the main grain production areas of China [7]. The food

security in the main grain production areas could have a profound impact on the security of the whole country. Therefore, it is very important to have accurate evaluation of the agricultural drought risk to ensure future grain production security.

Generally, drought can be classified into four types: Meteorological drought, agricultural drought, hydrological drought, and social-economic drought [8]. Meteorological drought is a prerequisite for the other three droughts. In order to effectively prevent and mitigate the drought disasters, drought research has attracted the attention of the government and scholars. The research about the definition of drought index [9,10], the causes of drought [11–13], the evolution of drought characteristics [14,15], and the impacts of drought [16,17] has received great attention. In terms of drought indexes, the Palmer Drought Severity Index (PDSI), the Standardized Precipitation Index (SPI), and the Standardized Precipitation Evapotranspiration Index (SPEI) have been utilized widely. PDSI takes the changes in surface water balance into account, while the acquisition is difficult because of the complex calculation process and the limited time scale [18]. SPI is a representative index for drought analysis, with the advantages of simplified calculation and multi-scale characteristic [19]. On the basis of the SPI, SPEI was proposed by Vicente-Serrano et al., which integrates the sensitivity of demand for evaporation of the PDSI and includes simple calculation and the multiple space–time attribute of the SPI based on meteorological data [20]. Therefore, SPEI is more comprehensive and reasonable for this research.

Agricultural drought risk is the consequence of a combination of meteorological factors and the drought damage vulnerability of agricultural production system [21]. From the perspective of disaster formation mechanism, its destructiveness depends on the intensity of disaster-causing factors and the vulnerability of disaster-bearing bodies. For drought risk research, no effective measures have been found to change the occurrence and development process of disaster-causing factors, so reducing drought vulnerability has become an important way for drought risk management [22]. Owning to the increasing aggravation of agricultural drought risk, research on drought vulnerability has grown rapidly. Shahid et al. applied a meteorological drought risk assessment framework that incorporates hazards and vulnerability by introducing a systematic three-step methodology [23]. Cheng et al. screened out 17 evaluations indexes mainly from a socio-economic perspective in seven counties of Xiaogan city, China, applied AHP to determine the weights of the evaluation indexes, and assessed the drought vulnerability by the fuzzy comprehensive evaluation method [24]. Rajsekhar et al. introduced various socioeconomic factors for drought vulnerability assessments [25]. De Silva et al. applied a case study of flood and drought impact in a rural Sri Lankan Community to illustrate the socioeconomic vulnerability to disaster risk [26]. Hlalele et al. used a force-field project management technique for drought vulnerability analysis [27]. Ali et al. implemented a rigorous framework to quantify drought vulnerability and assessed the drought risk across Africa [28]. Zeng et al. applied a widely accepted conceptual model that emphasizes the combined role of drought hazard and agricultural drought vulnerability to conduct a spatial assessment of agricultural drought risk [29]. Linlin Fan et al. used Copulas to quantify the curve between drought and water scarcity in the Metropolitan Areas [30]. Wang et al. established an integrated drought risk model based on the relation curve of drought joint probabilities and drought losses of multi-hazard-affected bodies [31].

In summary, most of the previous studies have considered the hazard factor and socio-economic factors when assessing drought vulnerability [23–29]. Some studies performed on the vulnerability curve, which is more accurate due to qualifying of the relationship between crop yield and drought [30,31]. However, meteorological and social-economic data may be limited because of insufficient observational data. Thus, based on "information distribution" and "information diffusion", the technique of fuzzy information optimization processing was proposed by Huang and Moraga [32]. Information distribution and diffusion methods have better advantages for small sample risk analysis. By using the information distribution method, the boundary of the histogram can be blurred to make full use of the information of the transition boundary. It is more accurate than the traditional histogram method when estimating the probability of the small sample. Meanwhile, the information diffusion method is based on the molecular diffusion theory, which normalizes the sample information and

diffuses it to a certain extent, solving the problem of insufficient capacity for small sample events [33,34]. Based on the method of information distribution and diffusion, this paper regarded drought strength as the drought hazard and drought damage rates as the result of the hazard, and the vulnerability function between them was constructed to analyze drought risk. Furthermore, conditional probabilities under different drought risk stages were analyzed, for there were few researches on the conditional probability of drought risk ever since.

In view of this, this paper starts by describing the study area, data, and methodologies. Then, after the drought thresholds are redefined by correlation analysis, the vulnerability relationship between drought strength calculated by SPEI and agriculture drought damage is established by information distribution and diffusion methods, and the provincial drought risks in the main grain production areas of China are calculated. Lastly, a discussion is employed and the basic conclusions of the full text are given.

## 2. Materials and Methods

### 2.1. Materials

#### 2.1.1. Study Region

The major grain production areas of China are mainly distributed in 13 provinces, covering an area of approximately 3.82 million km², or 39.8% of China, lying between latitudes 24°26′ N and 53°04′ N and longitudes 97°12′ E and 134°43′ E. The 13 provinces include Liaoning, Hebei, Shandong, Jilin, Inner Mongolia, Jiangxi, Hunan, Henan, Hubei, Jiangsu, Anhui, Sichuan (including Chongqing), and Heilongjiang (shown in Figure 1). According to the State Administration of Grain, the main grain production areas contribute about 75% of the country's total output, and about 95% of the country's total grain increment. It is obvious that the food security in China's major grain-producing areas has a vital impact on national security. Therefore, it is very crucial to have accurate evaluation of the agricultural drought risk to ensure future grain production security in the major grain production areas.

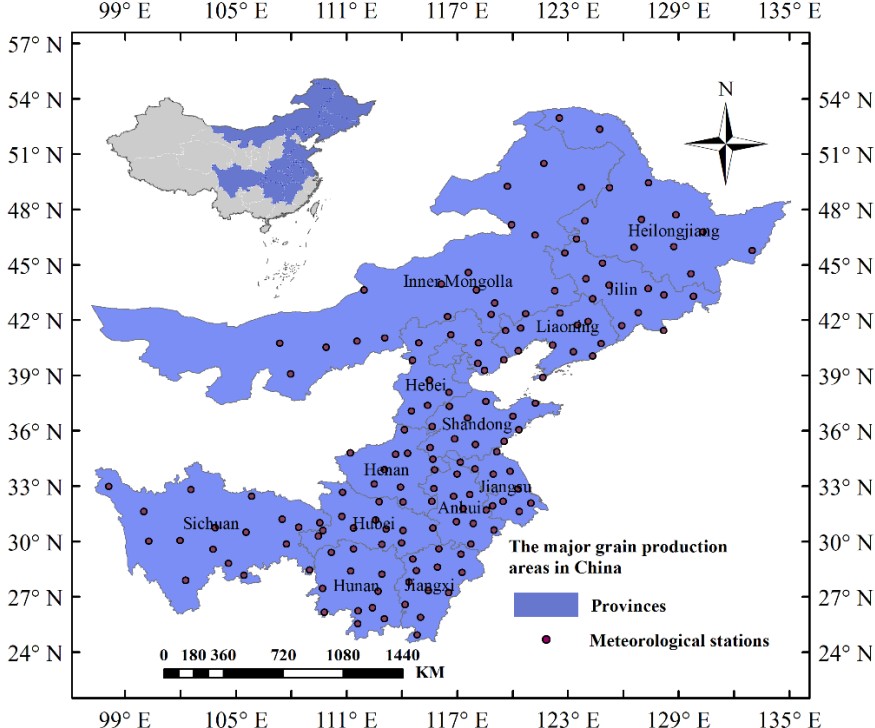

**Figure 1.** Location of the major grain production areas and the distributions of the meteorological stations and virtual stations.

According to the difference in planting structure, the study area can be divided into the main grain production areas of north China (including Hebei, Liaoning, Jilin, Heilongjiang, Shandong, Henan, and Inner Mongolia provinces) and the main grain production areas of south China (including Hubei, Jiangsu, Anhui, Sichuan Jiangxi, Hunan provinces). As shown in Figure 2, the grain output continued to increase from 1961 to 2017 in different regions and the national total grain output reached about 661 million tons in 2017. From the regional perspective, the proportion of grain yield in major grain production areas of China increased from 68.2% to 76.1%, where the proportion of grain yield in the north increased from 31.2% to 46.7%, while in the south, it decreased from 37.0% to 30.6%, to the national grain yield from 1961 to 2017. Figure 3 shows that in the main grain production areas of north China, wheat and corn were the main crops (yields counting for 30% and 53%, separately), while the grain yield ratio of rice, potatoes, and beans was relatively small (counting for 11%, 3%, and 3%, separately) in 2017. The ratio of corn and rice yield increased, while wheat, potatoes, and beans decreased from 1961 to 2017. In the main grain production areas of south China, rice played a dominant role (counting for 59%), while the ratio of rice yield decreased and the ratio of wheat and corn increased from 1961 to 2017.

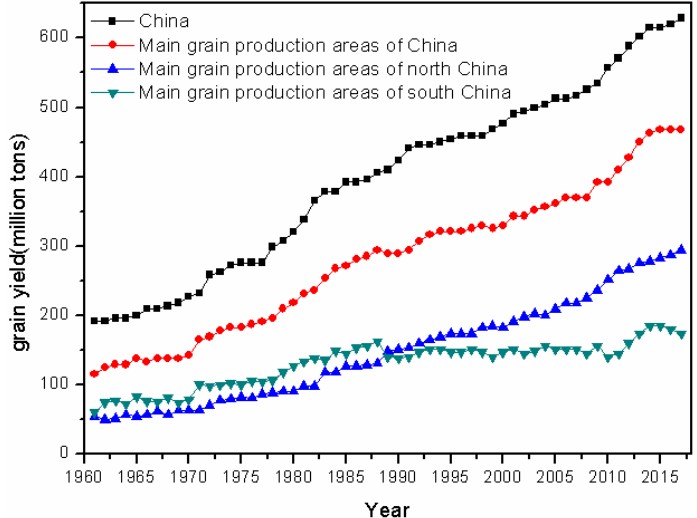

**Figure 2.** The trend of grain yield in different regions from 1961 to 2017.

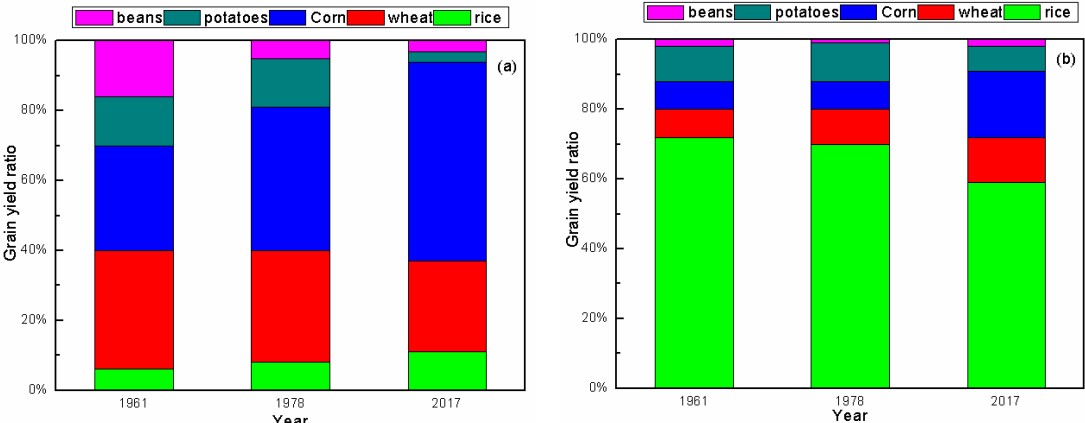

**Figure 3.** Changes of grain yield ratio in major grain production areas from 1961 to 2017. (**a**) The grain yield ratio in the major grain production areas of north China; (**b**) the grain yield ratio in the major grain production areas of south China.

### 2.1.2. Data

The monthly meteorological data used in this study were selected from 158 meteorological stations during 1961–2016 from the China Meteorological Data Sharing Service System (http://cdc.cma.gov.cn/home.do.), including precipitation, maximum air temperature, minimum air temperature, mean air temperature, relative humidity, wind speed, atmospheric pressure, and sunshine duration, where the missing data of the selected station are no more than 0.1%.

Other factors (breeding techniques, agricultural management techniques, pests and diseases, etc.) other than meteorological factors are related to drought damage. These factors are difficult to quantify, but could reflect on the crop yield eventually. So drought damage rates, including the drought-induced rate ($Y_1$), drought-affected rate ($Y_2$), and lost harvest rate ($Y_3$) of crops, can be used to evaluate the drought damage. From the perspective of agricultural disasters, the drought-induced area can be referred to as the area where crop yields reduced by more than 10% relative to the normal crop yields due to drought, floods, windstorms, frosts, pests, and other natural disasters during the year. When several disasters are suffered, the drought-induced area cannot be double-counted and should be calculated only according to the largest and most severe damage. The drought-affected area can be referred to as the area where crop yields reduced by more than 30% relative to the normal crop yields when suffering from the above natural disasters. The lost harvest area refers to the area where crop yields reduced by more than 70% relative to the normal crop yields when suffering from the above natural disasters [35]. The drought-induced area, drought-affected area, grain production area data from 1961 to 2016 (lack of data in 1965, 1967–1970), and the lost harvest area data from 1983–2016 in the 13 provinces were obtained from Chinese Plantation Management Department Network (http://www.zzys.moa.gov.cn/).

### 2.2. Methods

#### 2.2.1. Drought Damage Index

The drought damage index can be expressed by the drought damage rate, which is a possibility at a certain drought stage, including drought-induced rate ($Y_1$), drought-affected rate ($Y_2$), and lost harvest rate ($Y_3$) [36]. The calculation formulas were as follows:

$$Y_1 = \frac{A_1}{A} \tag{1}$$

$$Y_2 = \frac{A_2}{A} \tag{2}$$

$$Y_3 = \frac{A_3}{A} \tag{3}$$

where $A_1$, $A_2$, $A_3$, $A$ represent the drought-induced area, drought-affected area, lost harvest area, and grain production area, respectively.

#### 2.2.2. Drought Meteorological Index

The principle of the Standardized Precipitation Evapotranspiration Index (SPEI) uses the degree of the difference between precipitation and potential evapotranspiration (PET), which deviates the average status to represent the regional drought conditions. In this study, the SPEI was used to monitor and quantify the changes of drought conditions in the main grain production area of China.

The SPEI was developed by reference [20], and is a standardized value of the difference ($D_i$) of precipitation ($P_i$) and potential evapotranspiration (*PET*) for month *i*:

$$D_i = P_i - PET_i \tag{4}$$

Our Estimation of PET is based on the Hargreaves (Hg) equation [37]. The Hg equation requires monthly temperatures (maximum and minimum), monthly mean precipitation, and the latitudinal position of the site. Hargreaves (Hg) equation [37] showed that the use of the Hg method is the best option, especially in regions with limited data. The $D_i$ values can be aggregated over different timescales; however, the $D_i$ values are first standardized from a three-parameter log–logistic distribution.

$$f(x) = \frac{\beta}{\alpha} \left( \frac{x-r}{\alpha} \right)^{\beta-1} \left( 1 + \left( \frac{x-r}{\alpha} \right)^{\beta} \right)$$ (5)

The parameters $\alpha$, $\beta$, and $r$ are calculated from probability weighted moments (PWM) and used to determine the log–logistic distribution, which is applied to the $D_i$ data set. The SPEI is then calculated as the probability of exceeding a given value of $D_i$:

$$SPEI = W - \frac{C_0 + C_1 W + C_2 W^2}{1 + D_1 W + D_2 W^2 + D_3 W^3}$$ (6)

$$W = \sqrt{-2 \ln(P)} \text{ for } p \leq 0.5$$ (7)

where $p$ is the probability of exceeding a determined $D$ value, $p = 1 - F(x)$. If $p > 0.5$, $p$ is replaced by $1-p$ and the sign of the resultant *SPEI* is reversed. And the constants are $C_0 = 2.515517$, $C_1 = 0.802853$, $C_2 = 0.010328$, $D_1 = 1.432788$, $D_2 = 0.189269$, and $D_3 = 0.001308$. Positive values of SPEI indicate the above average moisture conditions, while negative values indicate the drier conditions [38]. Among them, the drought severity was classified according to the value of SPEI, shown in Table 1.

**Table 1.** The Standardized Precipitation Evapotranspiration Index (SPEI) grade standard divided for drought.

| Grade | Type | SPEI Value |
|-------|------|------------|
| 0 | Normal | >−0.5 |
| 1 | Mild drought | (−1.00, −0.5) |
| 2 | Moderate drought | (−1.5, −1) |
| 3 | Severe drought | (−2.0, −1.5) |
| 4 | Extreme drought | <−2.0 |

Crops in different growth periods have different sensitivity to drought, so it is necessary to select appropriate timescales for monitoring a certain drought. Considering SPEI of different timescales from January 1961 to December 2016, taking six timescales into account, including SPEI1 (1-month), SPEI3 (3-month), SPEI6 (6-month), SPEI9 (9-month), SPEI12 (12-month), and SPEI24 (24-month). The Tyson Polygon method, proposed by Dutch meteorologist Thiessen [39], initially used to calculate the average area rainfall of discretely distributed weather stations, can be applied to calculate the average SPEI value of the selected 13 provinces. The SPEI in each province can be calculated as follows:

$$X_a = \sum_{b=1}^{n} f_{ab} x_{ab}$$ (8)

where $X_i$ is the average SPEI value in the a-th province, $x_{ab}$ is the SPEI value of the b-th station in the a-th province, $f_{ab}$ is the proportion of the area of each Tyson polygon to the area of the province.

The values of drought strength can be calculated by Formula (9):

$$F = -\sum_{a=1}^{D} x_a$$ (9)

where *F* is drought strength in the province, *D* is the number of months when SPEI values are below the drought thresholds, namely drought duration. Traditionally, the range of drought threshold is determined by the specific value, which only considers meteorological factors and ignores actual disaster losses when defining drought levels. Therefore, the drought threshold was redefined in this paper. At first, the correlation coefficients between *F/D* and $Y_1$, $Y_2$, $Y_3$ are calculated, respectively, when S = [−2.0, −1.9, . . . . . . , −0.1, 0]. Then the drought threshold can be selected corresponding to the maximum correlation coefficient [40].

2.2.3. Information Distribution and Diffusion Methods

Effective Learning of Samples is one of the Key Aspects of Risk Analysis

In many cases, there are very few samples available, which is called incomplete information. Under this condition, the reliability of the results obtained by the traditional probability and relevant risk analysis conclusions cannot be guaranteed. Fortunately, based on "information distribution" and "information diffusion", the technique of fuzzy information optimization processing was proposed by Huang and Moraga [32]. Compared with the traditional histogram method, the information distribution method can be applied to estimate related real facts, expanding an observation into a fuzzy set, so as to optimally process a small sample, which can make the result more precise. Huang initially utilized information distribution and diffusion methods to estimate the risk of the annual flood disaster. In the study, the information distribution method was applied to estimate the probability distribution, and the normal information diffusion method was used to construct the vulnerability curve. By multiplying the probability distribution and the vulnerability curve and integrating them, the risk of the annual flood disaster in the study area can be calculated. The connotation can be interpreted as the expected value of the proportion of the affected population [41]. The detailed computational process of these methods was illustrated in the following the references [32,35,36,41]. The specific process of this method is as follows:

The observation data of X = {$x_1$, $x_2$, . . . , $x_n$} is regarded as the sample. The appropriate interval length $\Delta$ is obtained based on the maximum and minimum values in the sample, and the space of monitoring points U = {$u_1$, $u_2$, $u_3$, . . . , $u_m$} corresponding to the sample were generated. The information $q_{ij}$ carried by the point $x_i$ in the X is distributed to its corresponding control point space $u_j$ by information distribution. The formulas are as follows (10)–(14):

$$\Delta = \frac{f_{\max} - f_{min}}{1.87 \times (n-1)^{2/5}} \tag{10}$$

$$q_{ij} = \begin{cases} 1 - \frac{|x_i - u_j|}{\Delta}, |x_i - u_j| \leq \Delta, i = 1, 2, \ldots, n; j = 1, 2, \ldots, m \\ 0, others \end{cases} \tag{11}$$

Equation (10) is an empirical equation, where n is the sample number, $f_{max}$ and $f_{min}$ are the maximum and minimum values in the sample, $\Delta$ is the interval length, and $q_{ij}$ is the information carried by the point $x_i$ in the X distributed to its corresponding control point space $u_j$. The distribution information $Q_j$ can be obtained from the monitoring points $u_j$.

Let:

$$Q_j = \sum_{i=1}^{n} q_{ij} \tag{12}$$

All distribution information $Qj$ can be obtained from the monitoring point uj, which is called the primary information distribution of F on U.

$$p_j = \frac{Q_j}{n} \tag{13}$$

Therefore, we can employ Equation (10) to estimate the probability of a drought disaster in magnitude uj.

$$P = \{p(u_1), p(u_2), p(u_3), \ldots, p(u_m)\} = \{p_1, p_2, p_3, \ldots, p_m\} \tag{14}$$

Two-Dimensional Normal Information Diffusion Method

In the two-dimensional case, the meteorological element x and drought damage index y could be formed into a data set W, as shown in Equation (15):

$$W = \{(x_1, y_1), (x_2, y_2), \ldots, (x_n, y_n)\} \tag{15}$$

According to requirements, the interval lengths $\Delta_x$ and $\Delta_y$ are selected to generate monitoring space for input and output domain T and V, respectively

$$T = \{t_1, t_2, \ldots, t_m\}, \tag{16}$$

$$V = \{v_1, v_2, \ldots, v_t\} \tag{17}$$

The information quantity $\mu_{ijk}$ carried by the sample point $(x_i, y_i)$ in W can be assigned to the points in T and V by the two-dimensional normal diffusion equation. For any sample (x,y) in the sample set X, the normal distribution method can be used to spread to the input and output domain T, V.

$$\mu_{ijk} = \frac{1}{2\pi h_x h_y} \exp\left(-\frac{(x_i - t_j)^2}{2h_x^2} - \frac{(y_i - v_k)^2}{2h_y^2}\right) \tag{18}$$

where $h$ is named as normal diffusion coefficient, calculated by Equation (19):

$$h = \begin{cases} 0.8416(b-a) & m = 5 \\ 0.5690(b-a) & m = 6 \\ 0.4560(b-a) & m = 7 \\ 0.3860(b-a) & m = 8 \\ 0.3662(b-a) & m = 9 \\ 0.2986(b-a) & m = 10 \\ 2.6851(b-a)/(n-1) & m \geq 11 \end{cases} \tag{19}$$

where $b = \max\limits_{1 \leq i \leq n}\{x_i\}, a = \min\limits_{1 \leq i \leq n}\{x_i\}$ when calculating the coefficient $h_x$.

The original information matrix $Q$ can be calculated by Equations (20)–(22).

$$Q = \begin{cases} Q_{11} & \cdots & Q_{1t} \\ \vdots & \ddots & \vdots \\ Q_{m1} & \cdots & Q_{mt} \end{cases} \tag{20}$$

Similar to the one-dimensional case, the columns in Q are normalized to form a fuzzy relationship R, whose physical meaning is to characterize the fuzzy relationship between meteorological elements x and drought damage index y. According to the fuzzy approximate reasoning model, the corresponding output dependent variable y can be further generated by inputting the independent variable x. Thereby, a vulnerability function can be established.

$$\begin{cases} s_k = \max\limits_{1 \leq j \leq m} Q_{jk} \\ r_{jk} = Q_{jk}/s_k \\ R = \{r_{jk}\}_{m \times t} \end{cases} \tag{21}$$

The fuzzy information matrix R is the risk model constructed in this paper. The corresponding output dependent variable is then obtained according to different input arguments. If the input quantity is the determined value $x_0$, the corresponding fuzzy set $\widetilde{x_0}$, can be obtained according to Equation (22):

$$\mu_{x0}(t_j) = \begin{cases} 1 - |x_0 - t_j| / \Delta_x, |x_0 - t_j| \le \Delta; \\ 0, \quad others \end{cases} \tag{22}$$

The output fuzzy set $\widetilde{y_0}$ can be obtained by multiplying the fuzzy set $\widetilde{y_0}$ and the fuzzy matrix $R$.

$$\mu_{y_0}(v_k) = \sum_{1 \le j \le m} \mu_{x_0}(u_j) \times r_{jk} \qquad k = 1, 2, \ldots, t \tag{23}$$

A specific value $y_0$ can be obtained by substituting fuzzy set $\widetilde{y_0}$ into Equation (24).

$$y_0 = \frac{\sum\limits_{1 \le k \le t} \mu_{y_0}(v_k) \times v_k}{\sum\limits_{1 \le k \le t} \mu_{y_0}(v_k)} \tag{24}$$

Due to information distribution and two-dimensional diffusion method, the original small sample data are constructed into a fuzzy matrix that reflects its causal relationship, and then the different independent variables are substituted to obtain the corresponding independent variables, so that the causal relationship between the small samples becomes more precise for solving the problem of insufficient samples.

### 2.2.4. Vulnerability and Risk Evaluation

The evaluation of the risk of a drought event can be described as the form below:

$$R = H \circ D \tag{25}$$

where $R$ is the risk which can be expressed as the expected value of the loss, $H$ is the hazard-causing factor which is described by the "probability", $D$ is the vulnerability function of hazard-bearing bodies in the face of disasters which indicates the loss, and "$\circ$" is a synthesis rule indicating multiplication in this situation [41].

The risk can be calculated by the following equation:

$$R = \int p(x) \cdot f(x) dx \tag{26}$$

where $p(x)$ is the probability density function of the hazard-causing factor, and $f(x)$ is the hazard-vulnerability function of hazard-bearing body.

The risk can be expressed as below when its probability distribution is discrete:

$$R = \sum_{i=1}^{n} p(u_i) \cdot f(u_i) \tag{27}$$

### 2.2.5. The Conditional Probability of the Risk

If there are two events $A$, $B$, then $P(A|B) = P(AB)/P(B)$, the conditional probability is the probability that event $A$ will occur under the condition that event $B$ has already occurred. The conditional probability is expressed as $P(A|B)$. In this study, the probability of a drought-induced event can be expressed as $R_1$, the probability of a drought-affected event expressed as $R_2$, and the probability of a lost harvest event expressed as $R_3$. The conditional probability of drought risk mainly contains the three situations below:

(1)　The probability that a drought-affected event will occur under the condition that the drought-induced event has already occurred, expressed as $P(R_2/R_1)$:

$$P(R_2|R_1) = R_2/R_1 \qquad (28)$$

(2)　The probability that a lost harvest event will occur under the condition that the drought-induced event has already occurred, expressed as $P(R_3/R_1)$:

$$P(R_3|R_1) = R_3/R_1 \qquad (29)$$

(3)　The probability that a lost harvest event will occur under the condition that the drought-affected event has already occurred, expressed as $P(R_3/R_2)$:

$$P(R_3|R_2) = R_3/R_2 \qquad (30)$$

## 3. Results

### 3.1. Correlation Analysis Between Drought Strength and Drought Damage Rates

Correlation analysis was employed for choosing the most suitable timescales and thresholds of SPEI corresponding to the drought event in this study. Traditionally, the drought thresholds are determined by specific values (Table 1), which only considers meteorological factors and ignores actual disaster losses when defining drought stages. In this study, the drought disaster threshold S was redefined by the correlation between drought strength calculated by SPEI at different time scales and drought damage rates, which is more applicable to the study area for improving drought risk assessment. Therefore, the drought thresholds were redefined between meteorological factors and agricultural drought damage, which is more meaningful for actual agricultural production. The correlations between *F/D* and $Y_1$, $Y_2$, $Y_3$ were calculated respectively when S = [−2.0, −1.9, . . . , −0.1, 0], and the S can be selected as the drought threshold corresponding to the maximum correlation coefficient.

Taking Shandong province as an example, Figure 4 presents the correlation coefficients between *F/D* and $Y_1$, $Y_2$, $Y_3$ at the six timescales. *D* is the number of months when SPEI values were below the drought threshold. *F* is the accumulation of SPEI values which were lower than the threshold. It is shown that the correlation coefficients between *D* and $Y_1$, $Y_2$, $Y_3$ fluctuated evidently, as shown in Figure 4(b1–b3), while the correlation coefficients between *F* and $Y_1$, $Y_2$, $Y_3$ fluctuated relatively stably, as shown in Figure 2(a1–a3). *F* is the cumulative value of the SPEI values below the drought threshold. Furthermore, *F* can also reflect *D* to some extent, according to Equation (2). Therefore, choosing *F* as the index of meteorological factor was more appropriate.

Generally, the smaller the SPEI value is, the greater the probability of a severe drought will be. However, it can be seen that the correlation coefficients at different timescales showed an increasing tendency with the increase of S, as shown in Figure 4(a1–a3), which were not completely consistent with the general results. The major reason for this phenomenon is that drought strength was the cumulative value of SPEI, which was less than the threshold S. In other words, drought strength was not only related to the value of SPEI, but also with the number of SPEI values below S. In Figure 4(a1–a3), it is obvious that the correlation coefficients at the timescales of SPEI6, SPEI9, and SPEI12 were larger than those at other timescales. Therefore, to ensure the stability and reliability of the drought vulnerability characteristics with respect to risk, the timescales of SPEI6, SPEI9, and SPEI12 and the thresholds corresponding to the maximum correlations between drought strength and drought damage rates were selected for further analysis in Shandong province. At the timescales of SPEI6, SPEI9, and SPEI12, the maximum correlation coefficients between *F* and $Y_1$ were 0.49, 0.48, and 0.47, with the corresponding thresholds being −0.4, −0.1, and −0.6; the maximum correlation coefficients between *F* and $Y_2$ were 0.57, 0.59, and 0.59, with the corresponding thresholds being −0.4, −0.4, and −0.6; while the maximum

correlation coefficients between $F$ and $Y_3$ were 0.65, 0.70, and 0.68, with the corresponding thresholds being −1.2, −0.4, and −0.7, respectively.

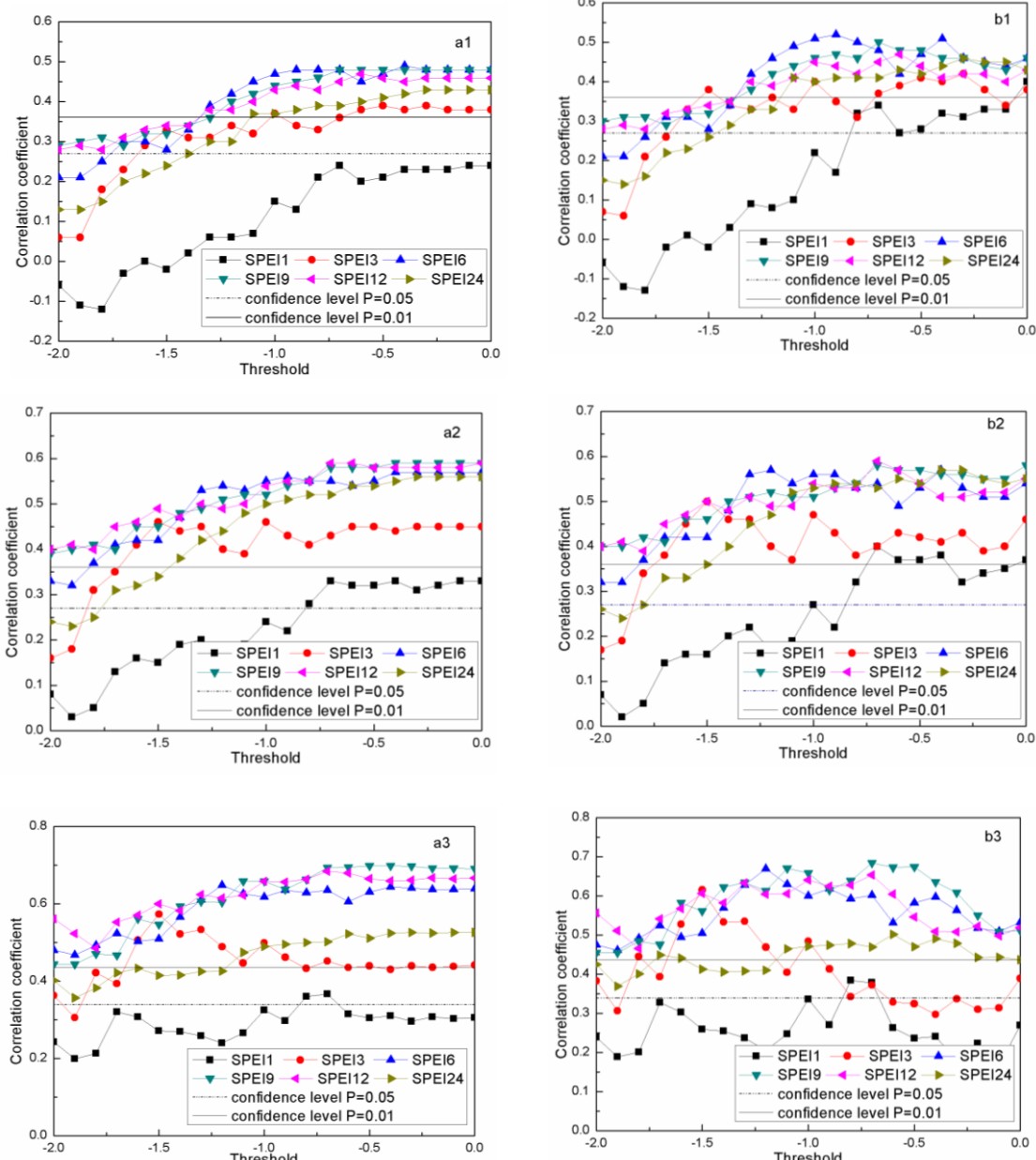

**Figure 4.** The correlation coefficient variations between drought strength, drought duration and drought-induced rate, drought-affected rate, and lost harvest rate in Shandong province at different timescales. (**a1**–**a3**) The correlation coefficient variations between drought strength and drought-induced rate, drought-affected rate, and lost harvest rate; (**b1**–**b3**) the correlation coefficient variations between drought duration and drought-induced rate, drought-affected rate, and lost harvest rate respectively. The broken line indicates a confidence level of $p$ = 0.05 and the solid line indicates a confidence level of $p$ = 0.01.

The maximum correlations and the corresponding thresholds can also be calculated in the other provinces according to the above-mentioned steps. The maximum correlation coefficients between $F$ and $Y_1$, $Y_2$, $Y_3$ at the top three timescales in each province were shown in Table 2 and the thresholds corresponding to maximum correlation coefficients are shown in Table 3. Conclusions can be drawn that the most suitable timescales, the maximum correlations, and the thresholds for monitoring drought



varied among the provinces, mainly due to the different cropping system and crop species across the study area. Most of the maximum correlations at the timescales of SPEI1, SPEI3, SPEI6, SPEI9, and SPEI12 passed a confidence level of $p = 0.05$, and even passed a confidence level of $p = 0.01$. However, only the correlations between $F$ and $Y_1$ of Heilongjiang and Henan provinces were not satisfied; the maximum correlation coefficients did not pass the confidence level of $p = 0.05$ at almost any timescales.

**Table 2.** The maximum correlations between drought strength and drought-induced rate, drought-affected rate, and lost harvest rate in the major grain production areas of China.

| Provinces | | SPEI1 | SPEI3 | SPEI6 | SPEI9 | SPEI12 | SPEI24 |
|---|---|---|---|---|---|---|---|
| Anhui | $CC_1$ | 0.48 ** | 0.60 ** | 0.64 ** | 0.60 ** | 0.46 ** | 0.16 |
| | $CC_2$ | 0.56 ** | 0.69 ** | 0.72 ** | 0.66 ** | 0.49 ** | 0.14 |
| | $CC_3$ | 0.49 ** | 0.57 ** | 0.60 ** | 0.54 ** | 0.42 ** | 0.36 ** |
| Hebei | $CC_1$ | 0.07 | 0.23 | 0.37 ** | 0.43 ** | 0.41 ** | 0.24 |
| | $CC_2$ | 0.36 * | 0.45 ** | 0.61 ** | 0.61 ** | 0.57 ** | 0.45 ** |
| | $CC_3$ | 0.30 * | 0.36 ** | 0.55 ** | 0.62 ** | 0.61 ** | 0.60 ** |
| Henan | $CC_1$ | 0.16 | 0.17 | 0.20 | 0.18 | 0.18 | −0.06 |
| | $CC_2$ | 0.37 ** | 0.38 ** | 0.44 ** | 0.41 ** | 0.36 * | −0.02 |
| | $CC_3$ | 0.31 * | 0.30 * | 0.31 * | 0.28 * | 0.27 * | 0.09 |
| Heilongjiang | $CC_1$ | 0.17 | 0.35 * | 0.25 | −0.11 | 0.19 | 0.18 |
| | $CC_2$ | 0.28 ** | 0.42 ** | 0.32 * | 0.02 | 0.24 | 0.24 |
| | $CC_3$ | 0.16 | 0.36 * | 0.25 | 0.36 * | 0.15 | 0.15 |
| Hubei | $CC_1$ | 0.57 ** | 0.57 ** | 0.55 ** | 0.47 ** | 0.35 * | 0.15 |
| | $CC_2$ | 0.60 ** | 0.51 ** | 0.54 ** | 0.42 ** | 0.35 * | 0.21 |
| | $CC_3$ | 0.61 ** | 0.57 ** | 0.66 ** | 0.62 ** | 0.44 ** | 0.15 |
| Hunan | $CC_1$ | 0.45 ** | 0.44 ** | 0.35 * | 0.19 | 0.11 | 0.03 |
| | $CC_2$ | 0.34 ** | 0.40 ** | 0.29 * | 0.17 | 0.08 | 0.04 |
| | $CC_3$ | 0.37 ** | 0.36 ** | 0.18 | 0.04 | 0.01 | 0.09 |
| Jilin | $CC_1$ | 0.25 | 0.55 ** | 0.45 ** | 0.38 ** | 0.30 * | 0.31 * |
| | $CC_2$ | 0.24 | 0.61 ** | 0.48 ** | 0.48 ** | 0.41 ** | 0.35 * |
| | $CC_3$ | 0.27 * | 0.56 ** | 0.50 ** | 0.51 ** | 0.46 ** | 0.35 * |
| Jiangsu | $CC_1$ | 0.68 ** | 0.75 ** | 0.79 ** | 0.74 ** | 0.67 ** | 0.41 ** |
| | $CC_2$ | 0.63 ** | 0.59 ** | 0.59 ** | 0.47 ** | 0.35 * | 0.06 |
| | $CC_3$ | 0.60 ** | 0.64 ** | 0.76 ** | 0.69 ** | 0.60 ** | 0.60 ** |
| Jiangxi | $CC_1$ | 0.60 ** | 0.71 ** | 0.64 ** | 0.51 ** | 0.35 * | 0.04 |
| | $CC_2$ | 0.52 ** | 0.61 ** | 0.61 ** | 0.45 ** | 0.30 * | 0.01 |
| | $CC_3$ | 0.47 ** | 0.59 ** | 0.49 ** | 0.35 * | 0.21 | −0.04 |
| Liaoning | $CC_1$ | 0.44 ** | 0.68 ** | 0.76 ** | 0.72 ** | 0.67 ** | 0.50 ** |
| | $CC_2$ | 0.54 ** | 0.73 ** | 0.84 ** | 0.83 ** | 0.77 ** | 0.56 ** |
| | $CC_3$ | 0.55 ** | 0.82 ** | 0.89 ** | 0.81 ** | 0.81 ** | 0.69 ** |
| Inner Mongolia | $CC_1$ | 0.41 ** | 0.46 ** | 0.51 ** | 0.46 ** | 0.36 * | 0.30 * |
| | $CC_2$ | 0.31 ** | 0.41 ** | 0.55 ** | 0.51 ** | 0.42 ** | 0.34 ** |
| | $CC_3$ | 0.43 ** | 0.60 ** | 0.69 ** | 0.66 ** | 0.60 ** | 0.46 ** |
| Shandong | $CC_1$ | 0.24 | 0.39 ** | 0.49 ** | 0.48 ** | 0.47 ** | 0.43 ** |
| | $CC_2$ | 0.33 * | 0.46 ** | 0.57 ** | 0.59 ** | 0.59 ** | 0.56 ** |
| | $CC_3$ | 0.37 * | 0.57 ** | 0.65 ** | 0.70 ** | 0.68 ** | 0.53 ** |
| Sichuan | $CC_1$ | 0.23 | 0.32 * | 0.43 ** | 0.42 ** | 0.32 * | 0.08 |
| | $CC_2$ | 0.33 * | 0.36 * | 0.48 ** | 0.52 ** | 0.40 ** | 0.15 |
| | $CC_3$ | 0.61 ** | 0.57 ** | 0.66 ** | 0.62 ** | 0.44 ** | 0.15 |

Note: $CC_1$, $CC_2$, $CC_3$ represent the maximum correlations between F and $Y_1$, $Y_2$, $Y_3$ when it was under drought-induced, drought-affected, and lost harvest conditions at different timescales respectively. * The value which passed a confidence level of $p = 0.05$. ** The value which passed a confidence level of $p = 0.01$.

**Table 3.** The thresholds corresponding to the maximum correlations between drought strength and drought-induced rate, drought-affected rate, and lost harvest rate in the main grain production areas of China.

| Provinces | | SPEI1 | SPEI3 | SPEI6 | SPEI9 | SPEI12 | SPEI24 |
|---|---|---|---|---|---|---|---|
| Anhui | $S_1$ | | −1.5 | −0.9 | −1.1 | | |
| | $S_2$ | | −1.5 | −1.3 | −1.1 | | |
| | $S_3$ | | −1.5 | −1.3 | −1.4 | | |
| Hebei | $S_1$ | | | −0.6 | −0.8 | −0.8 | |
| | $S_2$ | | | −0.1 | −0.1 | −0.5 | |
| | $S_3$ | | | −0.1 | −0.1 | −0.7 | |
| Henan | $S_1$ | | | −0.6 | −0.8 | −0.8 | |
| | $S_2$ | | −0.1 | −0.1 | −0.5 | | |
| | $S_3$ | −2 | −0.7 | −0.5 | | | |
| Heilongjiang | $S_1$ | −1 | −2 | −0.1 | | | |
| | $S_2$ | −0.6 | −1.8 | −0.1 | | | |
| | $S_3$ | | | −0.1 | −0.1 | −0.7 | |
| Hubei | $S_1$ | −1.6 | −0.1 | −0.7 | | | |
| | $S_2$ | −1.6 | −0.7 | −1.3 | | | |
| | $S_3$ | −1.6 | −1.6 | −1.6 | | | |
| Hunan | $S_1$ | −0.2 | −0.3 | −0.1 | | | |
| | $S_2$ | −0.2 | −1 | −0.1 | | | |
| | $S_3$ | −0.2 | −0.8 | | | | |
| Jilin | $S_1$ | | −1.8 | −0.5 | −0.1 | | |
| | $S_2$ | | −1.9 | −0.5 | −0.4 | | |
| | $S_3$ | | −1.8 | −1 | −0.6 | | |
| Jiangsu | $S_1$ | | −1.8 | −1.4 | −1.9 | | |
| | $S_2$ | −0.6 | −1.4 | −1.4 | | | |
| | $S_3$ | | −1.8 | −1.8 | −1.5 | | |
| Jiangxi | $S_1$ | −0.6 | −0.8 | −0.1 | | | |
| | $S_2$ | −0.7 | −0.8 | −1.9 | | | |
| | $S_3$ | −1.7 | −0.7 | −1.9 | | | |
| Liaoning | $S_1$ | | −0.6 | −0.1 | −0.1 | | |
| | $S_2$ | | −0.8 | −0.2 | −0.1 | | |
| | $S_3$ | | −1.6 | −1.7 | −1.7 | | |
| Inner Mongolia | $S_1$ | | −0.4 | −0.4 | −0.3 | | |
| | $S_2$ | | | −0.1 | −0.1 | −0.1 | |
| | $S_3$ | | | −1.5 | −1.5 | −1.5 | |
| Shandong | $S_1$ | | | −0.4 | −0.1 | −0.6 | |
| | $S_2$ | | | −0.4 | −0.4 | −0.6 | |
| | $S_3$ | | | −1.2 | −0.4 | −0.7 | |
| Sichuan | $S_1$ | | −0.6 | −1 | −1 | | |
| | $S_2$ | | −0.6 | −1 | −1.6 | | |
| | $S_3$ | −1.6 | | −1.6 | −1.6 | | |

Note: $S_1$, $S_2$, $S_3$ represent the thresholds when it was under drought-induce, drought-affected, and lost harvest conditions, respectively.

Due to the diversity of the disaster-caused factors and risk-bearing bodies, the relationship between $F$ and $Y_1$, $Y_2$, $Y_3$ could not be analyzed by a simple regression method, but should be analyzed by a nonlinear analysis method. Meanwhile, the calculation sequences were short, as the length was only 50. Hence, the information distribution and the two-dimensional normal information diffusion method were selected to analyze the vulnerability relationship between $F$ and $Y_1$, $Y_2$, $Y_3$.

*3.2. The Vulnerability Curve Between Drought Strength and Drought Damage Rates*

In this study, the information distribution and the two-dimensional normal information diffusion method were employed to construct the vulnerability relationship between $F$ and $Y_1$, $Y_2$, $Y_3$. Take Shandong province for example, at the timescale of SPEI6, the $F$ from 1961 to 2016 (except for 1965, 1967–1970) was calculated as follows: $F = \{q_1, \ldots, q_{50}\} = \{0.827, \ldots, 0.000\}$, according to Equation (9). By Equation (10), the interval length $\Delta$ was obtained as 1.95. In order to blur the boundary of the histogram, the $F$ was sorted in ascending order, and a control point was added at each end of the histogram. Therefore, the starting point of the histogram was 0 and the end point was 19.5, with the intervals 1.95. Thenm the frequency of the control point of $F$ can be obtained according to Equations (11)–(14). The frequency can be regarded as probability, where P = {P $(u_1)$, P $(u_2)$, P $(u_3)$, ⋯, P $(u_{11})$} = {0.2343, 0.2029, 0.1389, ... , 0.0183}. For the number of monitoring points in information diffusion, the results exhibited little difference between 50, 100, 150, and 200 through calculation, respectively. However, the more monitoring intervals and monitoring points there are, the greater the calculation complexity is. Therefore, 50 monitoring intervals and 51 monitoring points were properly to be selected for further calculation. To reflect the relationship between $F$ and $Y_1$, the information matrix Q and the normalized information matrix R were calculated according to Equations (15)–(20). Corresponding to the input value $\{u_1, u_2, \ldots, u_{11}\}$, the output values $(y_1, y_2, \ldots, y_{11})$ were obtained according to Equations (21)–(24), where Y ={y $(u_1)$, y $(u_2)$, y $(u_3)$,⋯, y $(u_{11})$} = {0.2332, 0.2107, 0.2007, ... , 0.3461}. Then, the 11 discrete points can be connected to a vulnerability curve, which reflects the drought vulnerability relationship between $F$ and $Y_1$.

The vulnerability curves of $F$ and $Y_1$, $Y_2$, $Y_3$ at the timescales of SPEI6, SPEI9, and SPEI12 were illustrated in Shandong province, respectively (Figure 5). It is obvious that the vulnerability lines were all nonlinear. The distinction between the vulnerability characteristics of the same drought damage rates at different timescales was slight, indicating that the vulnerability results in this study were reliable. The vulnerability lines displayed that $Y_1$, $Y_2$, $Y_3$ increased rapidly with the increasing of $F$ at the beginning, and after a small fluctuation locally, they no longer increased significantly and tended to be relative stable. This indicates that as the continuous aggravation of meteorological drought increased, the drought damage did not continuously increase significantly and tended to be relatively stable gradually, which was almost consistent with the realistic situation [35,42]. Some drought-resistance measures were adopted to resist the spread of drought, including storing water through agriculture water conservancy projects or utilizing precipitation forecasts to prevent the occurrence of agriculture drought. Likewise, the vulnerability curves of other provinces in the main grain production areas of China can also be obtained using the information distribution and the two-dimensional normal information diffusion methods. The vulnerability curves exhibited roughly similar variation trends, while they also appeared slightly difference due to their unique drought-causing factors and drought-bearing bodies among the provinces. Meanwhile, as the meteorological data are predictable, the drought damage rates corresponding to the meteorological conditions can be obtained by substituting the drought strength into the information matrix $R_{U^*V}$, which can be used as an annual forecast of drought risk to some extent.

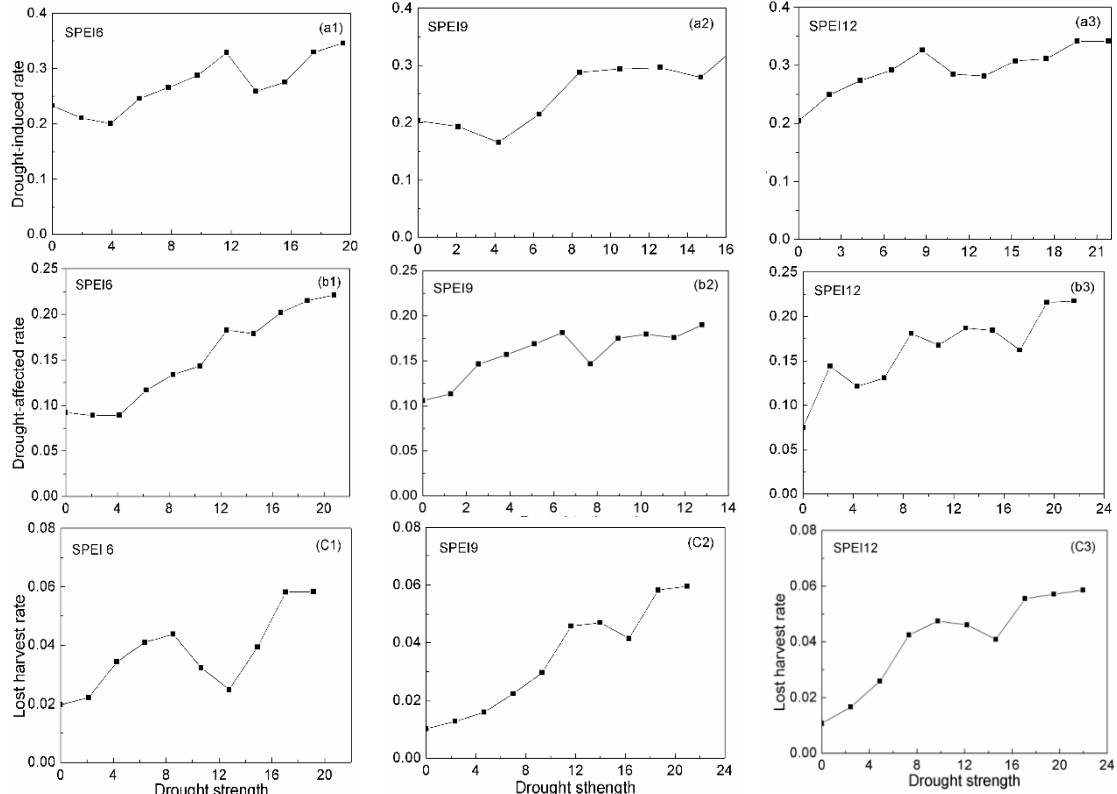

**Figure 5.** The vulnerability curve between drought strength and drought-induced rate, drought-affected rate, and lost harvest rate in Shandong province. (**a1**–**a3**) The vulnerability curve between drought strength calculated by SPEI6, SPEI9, and SPEI12 and drought-induced rate, respectively; (**b1**–**b3**) the vulnerability curve between drought strength calculated by SPEI6, SPEI9, and SPEI12 and drought-affected rate, respectively; (**c1**–**c3**) the vulnerability curve between drought strength calculated by SPEI6, SPEI9, and SPEI12 and lost harvest rate, respectively.

### 3.3. Agriculture Drought Risk Analysis

The provincial agriculture drought risks under different drought damage stages in the major grain production areas of China are shown in Table 4. The risk values were determined by Equations (27), where $P(u_i)$ was obtained by Equation (14), and $F(u_i)$ was the output value obtained by Equation (24). The drought risk can be understood as the percentage of loss from such drought events in the province.

The average provincial risk values of $R_1$, $R_2$, $R_3$ can be obtained by averaging the drought risk values at different timescales, for little differences among the risk values at the selected SPEI timescales (Table 4). It displayed that the three provinces with the highest drought-induced risk were Inner Mongolia, Jilin, and Liaoning, with the $R_1$ being 0.351, 0.346, and 0.309, while the three provinces with the lowest drought-induced risk were Hubei, Hunan, and Jiangxi, with the $R_1$ being 0.157, 0.156, and 0.093, respectively. The $R_2$ in Jilin, Inner Mongolia, and Liaoning were 0.217, 0.195, and 0.149, which were higher than those in other provinces, while Hunan, Jiangsu, and Jiangxi were 0.089, 0.066, and 0.063, which were lower than others. The $R_3$ in Inner Mongolia had the maximum value of 0.081, followed by Liaoning of 0.066, and Heilongjiang of 0.037, while the $R_3$ in Jiangxi had the minimum value of 0.009, followed by Jiangsu of 0.016, and Hunan of 0.022.

Figure 6 shows that the spatial distribution characters of $R_1$, $R_2$, and $R_3$ among the provinces were nearly identical. The provincial drought risks presented great spatial differences, with the common spatial distribution characteristics of high risk in the northern, moderate in the central part, and low under different drought damage rates in the southern provinces, which were consistent with previous research. Xie et al. employed one-dimensional a normal information diffusion method to identify drought disaster risk in China's major grain-producing areas and obtained provincial drought disaster

risk spatial distribution. They concluded that the agricultural drought risk was high in the northern and central parts of China, including Inner Mongolia, Heilongjiang, Jilin, Liaoning, and Hebei, while the agricultural drought disaster risk was lower in the southern provinces of Jiangxi, Jiangsu, Sichuan, and Anhui provinces, which is identical to our analysis conclusions in general [22]. Lu et al. found that the distribution of China drought risk showed a pattern of high in center, and the north areas higher than the south, increased gradually from southwest to northeast [43]. So the results obtained in the research can be trusted.

**Table 4.** Agriculture drought risk values based on different time scales for SPEI in the major grain production areas of China.

| | | SPEI1 | SPEI3 | SPEI6 | SPEI9 | SPEI12 | SPEI24 | Average | Rank |
|---|---|---|---|---|---|---|---|---|---|
| Anhui | $R_1$ | | 0.178 | 0.170 | 0.267 | | | 0.205 | 9 |
| | $R_2$ | | 0.117 | 0.102 | 0.137 | | | 0.118 | 9 |
| | $R_3$ | | 0.019 | 0.046 | 0.023 | | | 0.028 | 5 |
| Hebei | $R_1$ | | | 0.251 | 0.250 | 0.250 | | 0.250 | 5 |
| | $R_2$ | | | 0.121 | 0.129 | 0.128 | | 0.126 | 7 |
| | $R_3$ | | | 0.019 | 0.022 | 0.025 | | 0.023 | 10 |
| Henan | $R_1$ | | | 0.212 | 0.211 | 0.226 | | 0.217 | 8 |
| | $R_2$ | | 0.200 | 0.119 | 0.126 | | | 0.148 | 4 |
| | $R_3$ | 0.026 | 0.023 | 0.027 | | | | 0.026 | 7 |
| Heilong Jiang | $R_1$ | 0.244 | 0.251 | 0.299 | | | | 0.265 | 4 |
| | $R_2$ | 0.120 | 0.132 | 0.134 | | | | 0.129 | 5 |
| | $R_3$ | | | 0.041 | 0.026 | 0.043 | | 0.037 | 3 |
| Hubei | $R_1$ | 0.148 | 0.156 | 0.167 | | | | 0.157 | 11 |
| | $R_2$ | 0.077 | 0.134 | 0.125 | | | | 0.112 | 10 |
| | $R_3$ | 0.016 | 0.022 | 0.038 | | | | 0.024 | 9 |
| Hunan | $R_1$ | 0.107 | 0.162 | 0.199 | | | | 0.156 | 12 |
| | $R_2$ | 0.033 | 0.097 | 0.136 | | | | 0.089 | 11 |
| | $R_3$ | 0.014 | 0.021 | 0.029 | | | | 0.022 | 11 |
| Jilin | $R_1$ | | 0.491 | 0.280 | 0.267 | | | 0.346 | 2 |
| | $R_2$ | | 0.351 | 0.152 | 0.149 | | | 0.217 | 1 |
| | $R_3$ | | 0.005 | 0.049 | 0.033 | | | 0.029 | 4 |
| Jiangsu | $R_1$ | | 0.152 | 0.157 | 0.414 | | | 0.240 | 7 |
| | $R_2$ | 0.047 | 0.074 | 0.078 | | | | 0.066 | 12 |
| | $R_3$ | | 0.008 | 0.011 | 0.008 | | | 0.009 | 13 |
| Jiangxi | $R_1$ | 0.089 | 0.096 | 0.093 | | | | 0.093 | 13 |
| | $R_2$ | 0.047 | 0.068 | 0.073 | | | | 0.063 | 13 |
| | $R_3$ | 0.015 | 0.014 | 0.019 | | | | 0.016 | 12 |
| Liaoning | $R_1$ | | 0.313 | 0.279 | 0.336 | | | 0.309 | 3 |
| | $R_2$ | | 0.127 | 0.142 | 0.179 | | | 0.149 | 3 |
| | $R_3$ | | 0.058 | 0.075 | 0.095 | | | 0.076 | 2 |
| Inner Mongolia | $R_1$ | | 0.384 | 0.334 | 0.334 | | | 0.351 | 1 |
| | $R_2$ | | | 0.168 | 0.208 | 0.208 | | 0.195 | 2 |
| | $R_3$ | | | 0.081 | 0.081 | 0.080 | | 0.081 | 1 |
| Shandong | $R_1$ | | | 0.240 | 0.228 | 0.254 | | 0.241 | 6 |
| | $R_2$ | | | 0.114 | 0.122 | 0.120 | | 0.119 | 8 |
| | $R_3$ | | | 0.027 | 0.021 | 0.025 | | 0.025 | 8 |
| Sichuan | $R_1$ | | 0.206 | 0.242 | 0.168 | | | 0.206 | 10 |
| | $R_2$ | | 0.119 | 0.134 | 0.125 | | | 0.126 | 6 |
| | $R_3$ | 0.017 | | 0.020 | 0.023 | | | 0.027 | 6 |

Note: $R_1$ represents the drought risk under the drought-induced condition, $R_2$ represents the drought risk under the drought-affected condition, and $R_3$ represents the drought risk under the lost harvest condition.

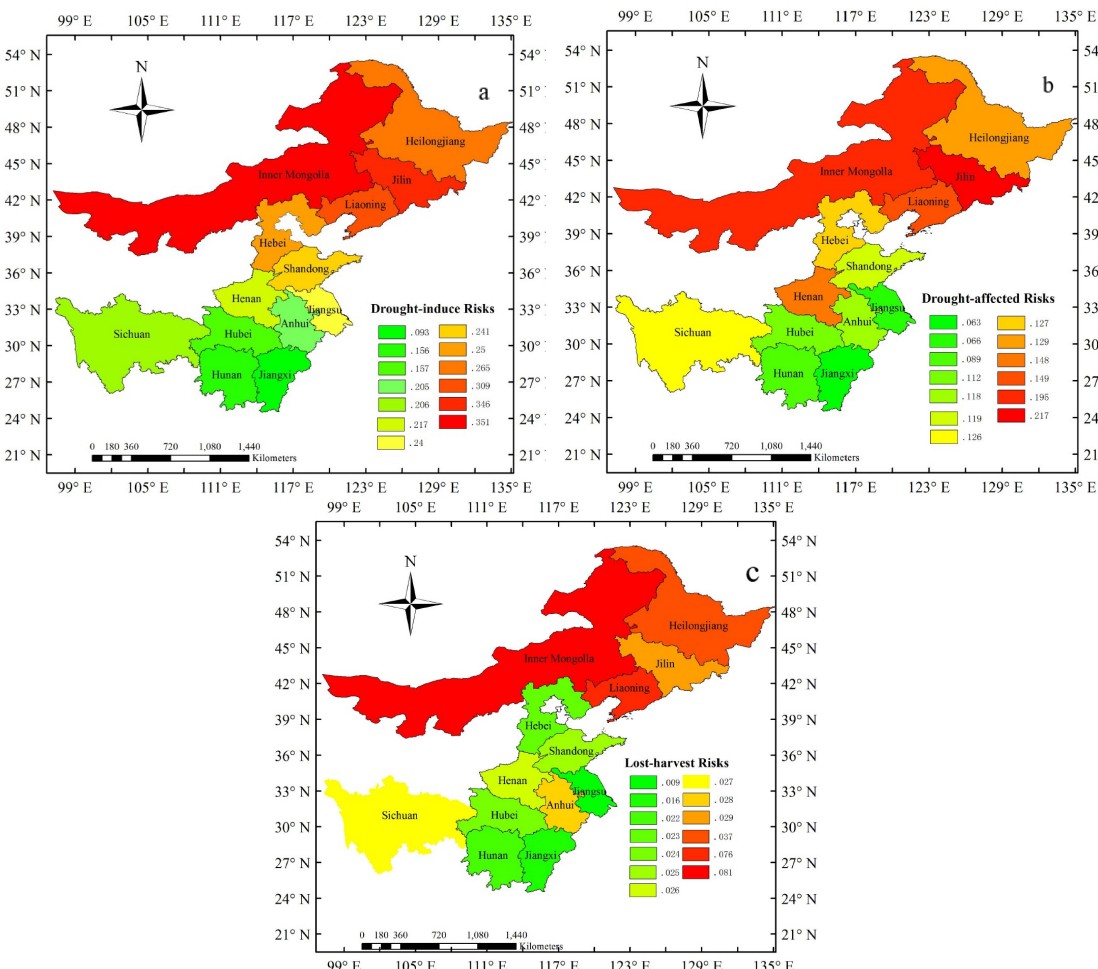

**Figure 6.** Provincial spatial distribution of the drought-induced risk, the drought-affected risk, and the lost harvest risk in the major grain production areas of China. (**a**) The provincial spatial distribution of the drought-induced risk; (**b**) the provincial spatial distribution of the drought-induced risk; (**c**) the provincial spatial distribution of the lost harvest risk.

### 3.4. The Conditional Probability of the Agricultural Drought Risk

The conditional probability values of $P(R_2|R_1)$, $P(R_3|R_1)$, and $P(R_3|R_2)$ in each province are shown in Table 5. The $P(R_2|R_1)$ in Hubei, Henan, and Jiangxi was 0.715, 0.684, and 0.677, which were higher than those in other provinces, while in Heilongjiang, Liaoning, and Jiangsu, it was 0.486, 0.483, and 0.275, which were lower than others. This means that when under drought-induced conditions, Hubei, Henan, and Jiangxi were the most prone to drought-affected disaster; however, Heilongjiang, Liaoning, and Jiangsu were the least prone to drought-affected disaster. The $P(R_3|R_1)$ in Liaoning had the maximum value of 0.512, followed by Inner Mongolia of 0.414, Hunan of 0.405, and Heilongjiang of 0.285, while the $P(R_3|R_1)$ in Jilin had the minimum value of 0.134, followed by Jiangsu of 0.136, Henan of 0.171, and Hebei of 0,173. Under the condition that drought-affected risk occurred, Liaoning, Hunan, and Inner Mongolia were the provinces most prone to lost harvest disaster, with the $P(R_3|R_2)$ being 0.247, 0.231, and 0.230, while Jiangsu, Jilin, and Hebei being the ones least to lost harvest disaster, with the $P(R_3|R_2)$ being 0.087, 0.084, and 0.037.

Figure 7 shows the provincial spatial distribution of the conditional probability values of $P(R_2|R_1)$, $P(R_3|R_1)$, and $P(R_3|R_2)$. The spatial distribution of the conditional probability values of $P(R_3|R_1)$ was almost consistent with that of $P(R_3|R_2)$, while far from that of $P(R_2|R_1)$.

**Table 5.** Provincial conditional probability of agriculture drought risk in the main grain production areas of China.

|  | $P(R_2|R_1)$ | $P(R_3|R_1)$ | $P(R_3|R_2)$ |
|---|---|---|---|
| Anhui | 0.578 | 0.249 | 0.144 |
| Hebei | 0.503 | 0.173 | 0.087 |
| Henan | 0.684 | 0.171 | 0.117 |
| Heilongjiang | 0.486 | 0.285 | 0.138 |
| Hubei | 0.715 | 0.226 | 0.162 |
| Hunan | 0.569 | 0.405 | 0.231 |
| Jilin | 0.628 | 0.134 | 0.084 |
| Jiangsu | 0.275 | 0.136 | 0.037 |
| Jiangxi | 0.677 | 0.257 | 0.174 |
| Liaoning | 0.483 | 0.512 | 0.247 |
| Inner Mongolia | 0.555 | 0.414 | 0.230 |
| Shandong | 0.493 | 0.205 | 0.101 |
| Sichuan | 0.613 | 0.217 | 0.133 |

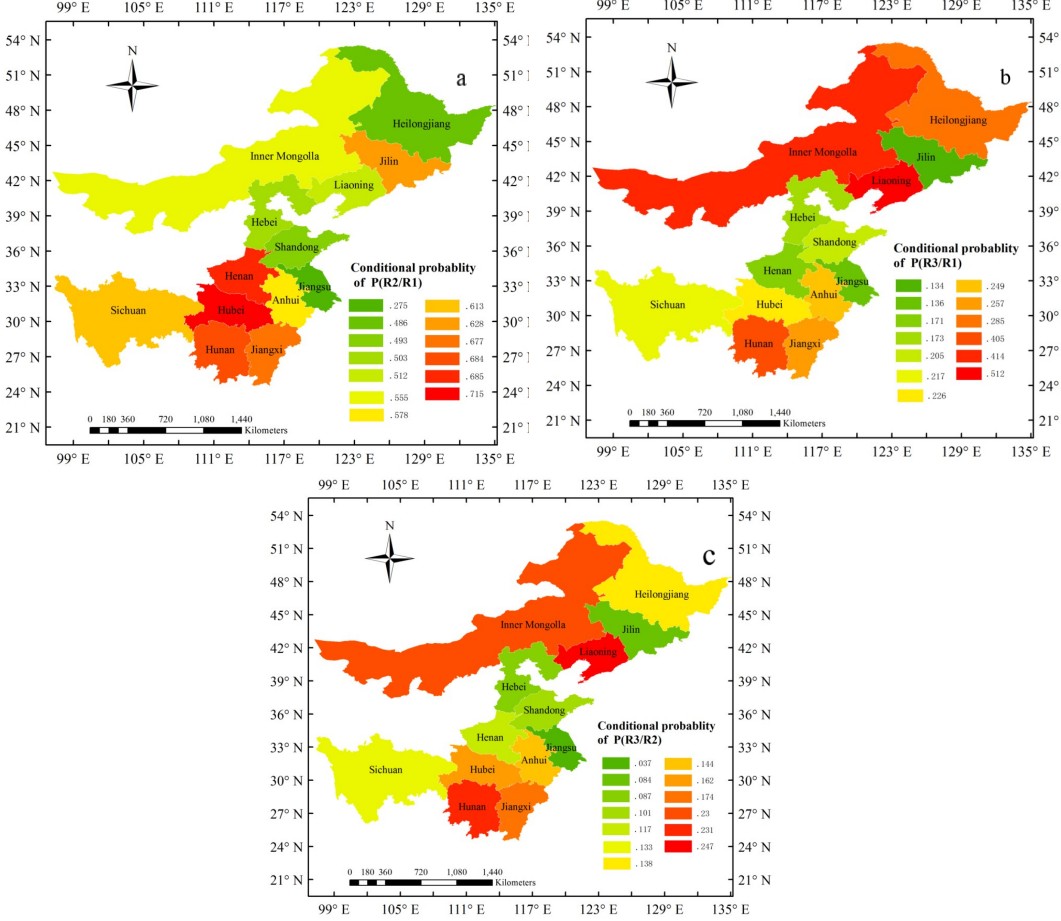

**Figure 7.** Provincial spatial distribution of the conditional probability values in the major grain production areas of China. (**a**) The provincial spatial distribution of the conditional probability values of $P(R_2|R_1)$; (**b**) the provincial spatial distribution of the conditional probability values of $P(R_3|R_1)$; (**c**) the provincial spatial distribution of the conditional probability values of $P(R_3|R_2)$.

## 4. Discussion

The correlations between $F$ and $Y_1$, $Y_2$, $Y_3$ at different timescales were quite different, presumably due to the differences in natural environment and the variety of crops planted among the provinces. Results showed that the correlations between $F$ and $Y_1$, $Y_2$, $Y_3$ were relatively high at the short time scales (1-month, 3-month, 6-month, 9-month, 12-month) in most provinces, but the correlation were relatively low for long timescale (24-month). The causes were that SPEI reflected the water content of vegetation and the soil moisture status at the short-term scale, which is closely related to agricultural production. SPEI at longer timescales, reflecting the reservoir's water storage capacity and groundwater level, has less impact on agriculture drought disasters, and may not be a proper index of agriculture drought conditions [34]. Therefore, the short timescales of SPEI selected for defining drought are reasonable in this study.

The provincial spatial distribution of agriculture drought risk in the major grain production areas of China is closely related to meteorological factors. As shown in Figure 8, the annual average precipitation and the annual average temperature values both gradually increased from north to south, with the common spatial distribution characteristics of low in the northern, moderate in the central part, and high in the southern provinces, which were opposite to the drought risk pattern. We also analyzed the linear relationship between provincial agricultural drought risk and meteorological factors in the major grain production areas of China (Figure 9). Agricultural drought risk (including drought-induced, drought-affected, and lost harvest risks) and meteorological factors (annual average precipitation and annual average temperature) all showed a negative relationship. Results showed that there were good linear relationships between drought-induced risk and annual average precipitation, and between drought-affected risk and annual average precipitation; the correlation coefficients were 0.6343 and 0.494, respectively. The relationship between lost harvest risk and annual average precipitation was not good, with the correlation coefficient being 0.2783. Meanwhile, the correlation coefficients between drought-induced risk, drought-affected risk, lost harvest risk, and annual average temperature were 0.6688, 0.4981, and 0.1197, respectively. Therefore, it can be concluded that precipitation and temperature have a great impact on drought when it is under drought-induced risk or drought-affected risk. As the drought continued to develop, the correlation coefficient became lower. Zeng ZQ et al. concluded that the main meteorological reason for the high frequency of extreme drought was the combined effect of abnormal decreases in rainfall and continued increases in temperature [29]. Wang HJ et al. indicated that drought in China was mainly concentrated around the changes in light and moderate drought and wind speed, precipitation, temperature were the most sensitive variable in northwest China, while in south China it was precipitation, followed by wind speed, temperature, relative humidity, and sunshine duration [44]. So we could infer that in the Chinese conditions, the lack of precipitation in the north has a stronger drought effect than the increase in temperatures towards the south. What is more, the difference of region economic development was also the cause of spatial diversity of drought risk. For backward economic and poor agricultural infrastructure in the northern and central provinces, the crops were mainly based on rain-fed agriculture, resulting in high drought risk. Meanwhile, in the southern provinces with developed economy, more irrigation facilities and advanced water management for preventing drought had relatively lower drought risk [45]. The spatial distribution of the conditional probability values was irregular, mainly because of the different drought-restricted strategies and measures adopted in different provinces. This can provide better assessment and monitoring of early stage of drought, which can help improve warning about preventing or mitigating the furthered adverse impacts of drought. Besides, the decision-making of the administrative departments is crucial in relation to drought prevention and management under climate change. Although drought may become more and more serious in the future, drought damage rates may not increase if more advanced irrigation facilities are constructed and water management applied.

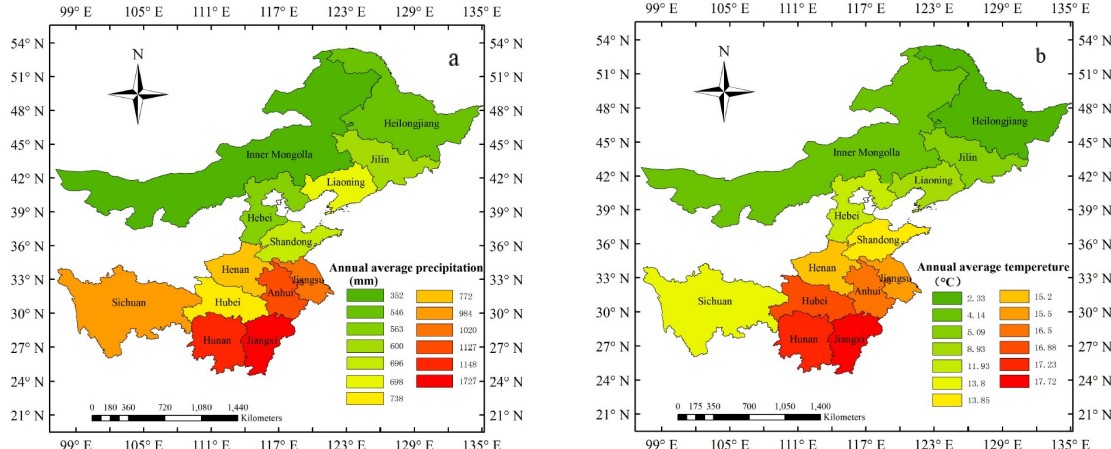

**Figure 8.** Provincial spatial distribution of the annual average precipitation and the annual average temperature in the major grain production areas of China. (**a**) The provincial spatial distribution of the annual average precipitation; (**b**) the provincial spatial distribution of the annual average temperature.

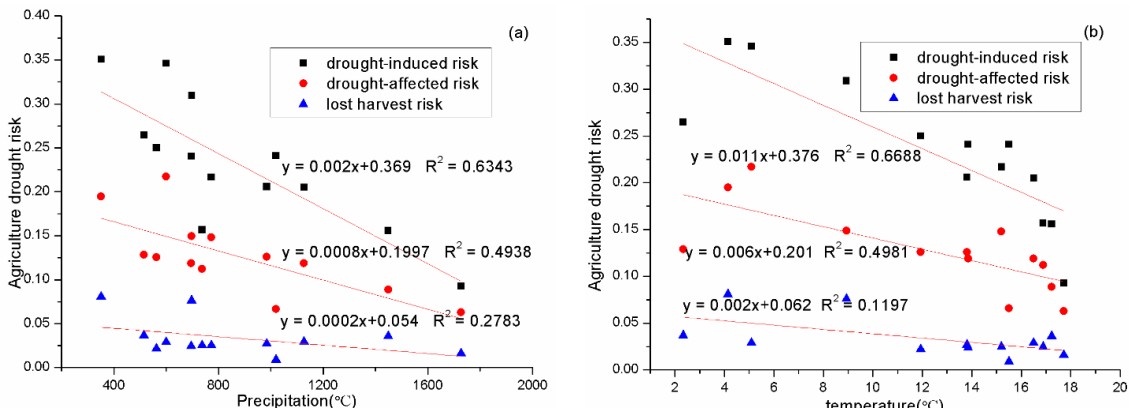

**Figure 9.** The linear relationship between provincial agricultural drought risk and meteorological factors in the major grain production areas of China. (**a**) The linear relationship between drought-induced risk, drought-affected risk, lost harvest risk, and the annual average precipitation; (**b**) the linear relationship between drought-induced risk, drought-affected risk, lost harvest riskm and the annual average temperature.

Agricultural drought hazard is the comprehensive reflection of many kinds of drought (i.e., meteorological, soil, hydrological), so it is difficult to accurately quantify agricultural drought risk by a single drought index. In recent years, there has been an increasing trend that emphasizes the combined role of drought hazard and drought vulnerability to conduct a risk assessment. For example, Zeng et al. assessed agricultural drought risk by combining the role of drought hazard (calculated by the intensity and frequency of drought) and agricultural drought vulnerability in southwest China [29]. David et al. presented a method for obtaining indices and maps of vulnerability to drought in Mexico; indices and maps were based on a set of socioeconomic and environmental indicators [46]. In the present study, we applied Standardized Precipitation Evapotranspiration Index (SPEI) as drought index to assess agricultural drought risk and combined drought hazard and agricultural drought vulnerability for agriculture drought risk evaluation, which was similar to previous studies like Zeng et al. in theoretical basis, that both drought hazard and drought vulnerability were combined to assess drought risk in southwest China. However, the methods of drought vulnerability calculation differed from each other. In the paper from Zeng et al. , agricultural drought vulnerability indexes were integrated with high-resolution soil properties, climate, topography, irrigation, and gross domestic product using

weights analysis, while the information distribution and the two-dimensional normal information diffusion methods were employed to establish the vulnerability curve between drought strength and drought damage rates in this paper. Information diffusion can compensate for the deficiency in sample information and can change a traditional data set into a fuzzy set by optimizing the use of the sample. Such an approach for agriculture drought risk assessment may be superior to traditional historical data-based methods because of the limited and insufficient observational data. What is more, if regional meteorological data can be predicted accurately by constructing the vulnerability curve, the drought damage rate can be estimated accordingly. As the vulnerable curves between drought strength and drought damage rates varied, along with the climate change and drought risk management levels, the current vulnerable curve should be sustained updated based on the latest data.

According to regional vulnerability characteristics, it can provide the basis for optimizing distribution of irrigation water resources and drought risk management, which would be beneficial to agricultural organizations, disaster management, and planning authorities. As the data are based on the provincial level, we can focus on the provincial level to provide scientific guidance for water resources and food management departments in different administrative regions. Besides, the average provincial risk values $(R_1, R_2, R_3)$ and the spatial distributions were obtained, which could provide guidance for the construction of national water transfer projects and water resources management in different areas. Furthermore, through conditional probabilities in each province, a reference for early warning of the likelihood of further aggravated drought can be provided. Nowadays, climate change impacts almost every part of our society—political, economic, ecological, social, cultural, technological, environmental, etc. As a consequence, there should be broad and integrated strategies for mitigating the impacts and adapting to climate change. Therefore, the socio-economic dimension should be an integral part of climate change discussion. Current literature on climate change is less than balanced among domains of scientific and human thought. This will probably change in the future, since the adaptation strategies are becoming an increasing concern in the scientific community. This article considered hazard factor and socio-economic factors on assessing drought vulnerability and it could provide suggestions that may be relevant for redefining policies aiming to improve water security at the main production areas of China. As the decision-making of the administrative departments is crucial in relation to drought prevention and management under climate change, drought damage rates may not increase if more advanced management is applied [47,48].

However, this article also has some shortcomings. Firstly, the provincial SPEI values were obtained by average SPEI of the sites using the Tyson polygon method. This may obliterate the characteristics of some sites and further produce some errors. We attribute these to the uncertainty of the research results. In addition, the normal diffusion function reflects a uniform diffusion process. However, most of the diffusion processes are asymmetric structures among the elements under practical applications. Therefore, it is necessary to consider the different diffusion velocity and modes at different directions, that is, the asymmetric diffusion of information for further study.

## 5. Conclusions

Agricultural drought is the primary disaster affecting grain production, resulting from meteorological drought and the vulnerability of the agricultural production system to meteorological drought. In this study, information distribution and the two-dimensional normal information diffusion methods were employed to establish the vulnerability curve between drought strength based on SPEI and agriculture drought damage, and then provincial drought risks and the conditional probabilities at different drought damage stages were obtained in the main grain production areas of China. The results showed that the drought vulnerability curve was nonlinear. With the increase of drought strength, drought damage rates increased rapidly at the beginning, and after a small fluctuation locally, they no longer increased significantly and tended to be relatively stable.

The provincial drought risk spatial distributions in the main grain production areas of China presented great differences, with the characteristics of high in the northern, moderate in the central

and southwestern, and lower in the southeastern provinces. The provincial drought risks revealed that Inner Mongolia, Jilin, and Liaoning were the provinces most prone to drought-induced risk, with the $R_1$ being 0.351, 0.346, and 0.309, while Hubei, Hunan, and Jiangxi were the ones least prone to drought-induced risk, with the $R_1$ being 0.157, 0.156, and 0.093, respectively. The $R_2$ in Jilin, Inner Mongolia, and Liaoning were 0.217, 0.195, and 0.149, which were higher than those in other provinces, while the $R_2$ was 0.089, 0.066, and 0.063 in Hunan, Jiangsu, and Jiangxi, which were lower than others. The $R_3$ in Inner Mongolia had the maximum value of 0.081, followed by Liaoning of 0.066, and Heilongjiang of 0.037, while the $R_3$ in Jiangxi had the minimum value of 0.009, followed by Jiangsu of 0.016, and Hunan of 0.022. Through cause analysis, it has been inferred that in the Chinese conditions, the lack of precipitation in the north has a stronger drought effect than the increase in temperatures towards the south. Besides, the difference of regional economic development was also the cause of spatial diversity of drought risk.

The conditional probability of drought risk showed that the $P(R_2|R_1)$ in Hubei, Henan, and Jiangxi was 0.715, 0.684, and 0.677, which were higher than those in other provinces, while in Heilongjiang, Liaoning, and Jiangsu, it was 0.486, 0.483, 0.275, which were lower than others. The $P(R_3|R_1)$ in Liaoning had the maximum value of 0.512, followed by Inner Mongolia at 0.414, Hunan at 0.405, and Heilongjiang at 0.285, while the $P(R_3|R_1)$ in Jilin had the minimum value of 0.134, followed by Jiangsu at 0.136, Henan at 0.171, and Hebei at 0,173. Under the condition that drought-affected risk occurred, Liaoning, Hunan, and Inner Mongolia were the provinces most prone to lost harvest risk, with the $P(R_3|R_2)$ being 0.247, 0.231, and 0.230, while Jiangsu, Jilin, and Hebei being the ones least prone to lost harvest risk, with the $P(R_3|R_2)$ being 0.087, 0.084, and 0.037. The spatial distribution of the conditional probability values was irregular, mainly because of the different drought-restricted strategies and measures adopted in different provinces. The results can provide better assessment and monitoring of the early stage of drought.

**Author Contributions:** All authors were involved in designing and discussing the study. K.N. and S.J. undertook the data analysis and drafted the manuscript. H.Y. collected the required data. L.Z. and Q.H. revised the manuscript and edited the language. C.L. and Y.W. contributed to the set-up of the simulations and the write-up of the paper. All authors read and approved the final manuscript.

**Funding:** This research was funded by the National Key Research and Development Program of China (No. 2016YFC0401508; 2016YFC0400902) and the Youth Program of National Natural Science Foundation of China (No. 51809252,51609140).

**Acknowledgments:** The authors would like to thank the National Climatic Centre of the China Meteorological Administration for providing the climate database used in this study. They also thank the anonymous reviewers and editors of this paper for their very helpful comments and suggestions.

**Conflicts of Interest:** The authors declare no conflicts of interest.

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
