# Peer review of "Analysis of Agricultural Drought Risk Based on Information Distribution and Diffusion Methods in the Main Grain Production Areas of China"

_atmosphere, doi:10.3390/atmos10120764_

Round 1

Reviewer 1 Report

Interesting analysis, however, I am not sure how it adds to what is already known about ET, Precip, crop type and irrigation across the provinces, and drought vulnerability. Main comments:

1.SPEI is introduced in section 2.2.1., however, it does not show how it is calculated. I understand it involves the difference between precip and ETpot, but how is it used to compute SPEI, and what does it mean whether it is positive or negative, and its magnitude.

2. A different places in the manuscript, it points to the paucity of data and the associated uncertainty of the presented analysis. However, as I understand from section 2.1.2 all needed data are available. It would be useful to explain what data are missing that are limited the analysis.

3. The theory section in 2.2.3 is very difficult to read, with the wide use of notation, yet some notations are not clear. What is m in Eq. 8, as it does not seem to be an integer as required when it denotes the nr of monitoring points (see also Eq. 18). What is S in line 268, fmax and fmin in Eq. 9, and so on. Instead, could this section be reduced and only focus on the main parameters that are used in the results and discussion, with using references about the details.

4. Regarding the results in section 2, for example lines 267-300 state that there is slight changes and eventual stability. This is difficult for me to accept, and certainly not very quantitative. I am wondering whether there is better interpretation to be gotten from Fig. 2.

The same can be said of the results presentation in section 2.1.2 (lines 330-335), when concluding that the curves in Fig. 4 are nonlinear. I really do not see this either.

5. Define R1, R2 and R3 as used in 2.1.3, in section 3.5

Author Response

Dear reviewer:

Thanks for your advice. We have revised according to your suggestions. Please see the attachment.

Reviewer 2 Report

A brief summary (one short paragraph) outlining the aim of the paper and its main contributions.

The aim of the paper is to find among the chinese regions which are more prone to drought and which are submitted to light (or short) or strong (or long) drought

Moreover, one of the conclusion could be about the relative effect of low precipitations or high temperatures : in the Chinese conditions, the lack of precipitations in the North has stronger drought effect than the increase in temperatures towards the South

Broad comments highlighting areas of strength and weakness. These comments should be specific enough for authors to be able to respond.

The paper is interesting, BUT after reading the whole paper, we are not sure that the tools are satisfactorily used, the best proof could be the pertinence of the final results, that is the consistency with the spatial distribution of the precipitations, more restrictive than temperature in the Chinese context.

Introduction bibliography

Even if the number of cited papers is high they are not sufficiently cited, -some are for the methods, other for the results, some are in small regions, other in big regions, in diverse continents (and climates)- and the knowledge they bring.

Moreover, the long series (like 23-31) of citations don't bring interesting information

l 100 You didn't explain why you need to redefine the drought thresholds. I suppose that it is mainly because of the high variability between the involved crops in the large range of the relevant climates.

The lack of justification underlines the need to have a brief description of the crops, their proportion and season in each province, when describing agricultural data. Probably, some have short cycles, while other extend over two calendar years. It could be interesting to link the results (thresholds, duration) with the main crops present in each province.

Moreover, it is not normal to mention thresholds before the basic (crucial) data, the agricultural data.

For the agricultural data (l 127-134), the presentation is very short, you didn't give any information on their spatial grain and their reliability.

Is it one value for one year, one crop and one province ?

The reader can't assess if the use you did of data is appropriate. You don't explain how was estimated the effect of drought ? In other words, are all the decreases of yield assigned to drought ? How were taken into account the effect of other limiting factors like pests and diseases ? Or what are the methods used to distinguish the factors reducing yields ?

For the calculation of SPEI, you content yourself with the citation of Vicente-Serrano, instead of explain the main hypothesis of this work, the only need of the temperature without radiation or wind values.

The methods of spatialisation of the stress criterion are well detailed, perhaps too lenghtily explained with details, while the hypotheses and the aim of these calculations are not sufficiently expressed. Is it better to spatialize SPEI than the initial meteorological criteria ? For the 1D and 2D methods, one or 2 small scheme to explain the distribution and diffusion methods could help to understand the principles of these methods, as they are coming from other disciplines.

Finally, is the aim to obtain one value per province for SPEI and drought indices ?

Without any comment on the meaning of the results, they don't seem realistic, but can only be the results of many calculations.

The presentation of the meteorological data only at the end, while they are inputs from the model is not convenient. The paper should be constructed on one question around the relative weight of precipitations and temperature to explain regional drought in China. In this perspective, it would be more interesting and demonstrative.

Specific comments

details to help understanding

l 122 data and not date

l 127 please 2 paragraphs, one for each type of data : meteorological and agricultural

Here, it could be positive to add some words on the crops (see above), like "rice in the South (2 or 3 in the year, with very short cycle duration), wheat in the North straddling 2 years" (I don't know)

Sometimes, the simple past is used when the sentence describes something permanent (and not past) : ex l 99-104

As the results are presented both in tables and figures for several situations (length of SPEI, province), they are easier to understand. This is a positive point in the global complexity of the paper.

l 78-87 :

References

The number of references are sometimes wrong, like Chen with n° 27 instead of 29 or Wang 32 instead of 35, Huang & Moraga 27 instead of 36

There is (at least) one lacking reference: Zhaoqi (l 420)

A complete control is to be done

l 134

The address of the the site on agricultural data does not work. Perhaps the true name is Chinese Agriculture (instead of planting) Information Network.

All these mistakes decrease reliability of the whole work.

l 182 and followings, eq (8) : no explanation of "m"

Author Response

(The authors gave the same response as above.)

Reviewer 3 Report

Authors worked hard on the study and I believe the outcomes can be considered. Yet there are rooms to improve substantially the paper. Some of the sentences are two longs and I recommend you to rephrase them. Remember “write simple, the better you communicate”. Formulas need attention and corrections. Please, find below my comments

1-) Do not start the title with “The”. “Analysis of agricultural drought… China” is better

2-) Line 27: insert “the” in front of drought-induced as follow “…risk under the drought-induced...”

3-) use these are keywords: agricultural drought; vulnerability; risk assessment; SPEI; information distribution; information diffusion; China

4-) Line 35-37: rephrase the sentence this way “Being one of the most reported natural disasters of the last decades, drought often causes severe damages to the society, the natural ecosystems, and the economy [1-3].”

5-) Line 69-70: make this change to the sentence “Owning to the increasing aggravation of agricultural drought risk, research on drought vulnerability has grown rapidly.

6-)  Report authors properly . E.g. Line 59  Vicente–Serrano et al. (2010) “ instead of “Vicente–Serrano et al in 2010 “;  line  70 “Hlalele (2019) used…” instead of  “Hlalele et al used”. Please check all these types of errors and correct the manuscript.

7-) Line 86: “the technique of fuzzy..” instead of “the technology of fuzzy”. Please check correction all over the document.

8-) Line 89: what do you mean by “fuzzifierred”??? Please rephrase your sentence and use appropriate words.

9-) 108 : Is this an actual fact? If yes, the tense should be present here “The major grain production areas of China are mainly distributed in 13 provinces..”

10-) Line 168-169: revise the sentence, it is too long and some grammar errors. e.g. “..were calculated respectively..” instead of “…was calculated respectively…”

11-) I recommend to strengthen your discussion section. You can read some useful article : https://doi.org/10.1016/j.jaridenv.2017.05.002   / DOI: http://dx.doi.org/10.18063/ESP.2016.01.002

12-) Line 154-156 “The Tyson Polygon method…” add reference(s).

13-) Please double check formula (6) and (7). In equation (6) you defined i as indicator of province (line 159-161); however in (7) you seem to define i as month 1≤i≤D with D=number of months (line 163). These need correction. I suggest that you review carefully all the formulas in the manuscripts and provide the meaning of the component of each equation.

14-) Notice: Understand that the curve of drought index values are generally tailed and that is a reason thresholds are used to define drought categories. Now, when you average SPEI values directly for a given province, the result  may not reflect the actual physical meaning.

15-) In Figure 4  please double check you Drought strength axes. First correct the label “Drought strength” for (c2) and (c3). Also I though your SPEI values should be negative 0 to negative infinite. See the axes of drought strength in figure 4 and make correction accordingly.

Author Response

(The authors gave the same response as above.)

Round 2

Reviewer 2 Report

I have 2 main negative comments.

First, and important enough, there is another paper with similar methods and results which is only cited at the end of this paper : the paper from Zeng et al (Water, 2019), while the present paper seems to be a spatial extension of the previous paper, from OsuthWest to the whole region of grain production.

Second, there are still like number 26 which is not "de SMMGT but De Silva MMGT.

Moreover, the paragraph concerning agricultural data seems not to have been corrected: the website address is wrong, I didn't find more escription on the main crops present in the studied area, or explanations on the assessment of the origin of the reduction of yields in these data: is ir strongly an effect of drought ?

I am annoyed to be obliged to refuse the paper in the present form

Author Response

(The authors gave the same response as above.)

Reviewer 3 Report

The paper can be considered for publication

Author Response

Dear reviewer:

Thank you very much!

Round 3

Reviewer 2 Report

I well understood the answer on the use (or not use) of the paper from Zeng et al (2019), but I require the mention of this paper in the Introduction (perhaps, I didn't say this sufficiently clearly in the previous comment) : I think that as  there are many other ideas than in the Zeng paper, it is better to show both the use of the Zeng paper and the innovations compared to this paper. In the paragrph of the introduction designing the methods, it will explain clearly the way used in the paper.

Thank you for correctingseveral references and adding a description of the data used. Nevertheless, even if I didn't verify all the references, bI don't trust them, as each reference I tried to use in the version 1 and 2 was wrong. Please test each.

Author Response

(The authors gave the same response as above.)
